# Intramuscular Fat Selection in Rabbits Modifies the Fatty Acid Composition of Muscle and Liver Tissues

**DOI:** 10.3390/ani12070893

**Published:** 2022-03-31

**Authors:** Agostina Zubiri-Gaitán, Agustín Blasco, Ruth Ccalta, Katy Satué, Pilar Hernández

**Affiliations:** 1Institute for Animal Science and Technology, Universitat Politècnica de València, 46022 Valencia, Spain; agzugai1@upvnet.upv.es (A.Z.-G.); ablasco@dca.upv.es (A.B.); ruthccaltahancco1992@gmail.com (R.C.); 2Department of Animal Medicine and Surgery, Universidad Cardenal Herrera-CEU, 46115 Moncada, Spain; ksatue@uchceu.es

**Keywords:** fat deposition, meat production, meat quality, divergent selection, plasma metabolites, liver fat, liver metabolism

## Abstract

**Simple Summary:**

Intramuscular fat content improves the juiciness, tenderness, and flavor of meat, but it can also affect its nutritional quality. A divergent selection experiment for intramuscular fat content was performed on rabbits for 10 generations to study the metabolism of the selected and correlated traits. The direct response to selection and the correlated responses in the meat fatty acid content, in the liver fat and its fatty acid content, and in plasma metabolic markers related to liver metabolism were studied. Increasing intramuscular fat content led to higher fat deposition in the carcass, but not in the liver. The fatty acid contents of *Longissimus thoracis et lumborum* muscle and liver were modified after selection, for which the microbiome composition also played an important role. A higher concentration of plasma lipids was found in the low-IMF line, probably due to a lower uptake by the muscle and adipose tissue.

**Abstract:**

This study was conducted on two rabbit lines divergently selected for intramuscular fat (IMF) content in the *Longissimus thoracis et lumborum* (*LTL*) muscle. The aim was to estimate the direct response to selection for IMF after 10 generations, and the correlated responses in carcass quality traits, meat fatty acid content, liver fat and its fatty acid content, and in plasma metabolic markers related to liver metabolism. Selection for IMF content was successful, showing a direct response equivalent to 3.8 SD of the trait after 10 generations. The high-IMF line (H) showed a greater dissectible fat percentage than the low-IMF line (L), with a relevant difference (D_H-L_ = 0.63%, P_r_ = 1). No difference was found in liver fat content (D_H-L_ = −0.04, P_0_ = 0.62). The fatty acid content of both *LTL* muscle and liver was modified after selection. The *LTL* muscle had greater saturated (SFA; D_H-L_ = 5.05, P_r_ = 1) and monounsaturated fatty acids (MUFA; D_H-L_ = 5.04, P_r_ = 1) contents in the H line than in the L line. No relevant difference was found in polyunsaturated fatty acids content (PUFA; P_r_ = 0.05); however, greater amounts of C18:2n6 (D_H-L_ = 3.03, P_r_ = 1) and C18:3n3 (D_H-L_ = 0.56, P_r_ = 1) were found in the H than in the L line. The liver presented greater MUFA (D_H-L_ = 1.46) and lower PUFA (D_H-L_ = −1.46) contents in the H than in the L line, but the difference was only relevant for MUFA (P_r_ = 0.86). The odd-chain saturated fatty acids C15:0 and C17:0 were more abundant in the liver of the L line than in the liver of the H line (D_H-L_ = −0.04, P_r_ = 0.98 for C15:0; D_H-L_ = −0.09, P_r_ = 0.92 for C17:0). Greater concentrations of plasma triglycerides (D_H-L_ = −34) and cholesterol (D_H-L_ = −3.85) were found in the L than in the H line, together with greater plasma concentration of bile acids (D_H-L_ = −2.13). Nonetheless, the difference was only relevant for triglycerides (P_r_ = 0.98).

## 1. Introduction

Intramuscular fat (IMF) is one of the main parameters influencing meat quality, and its importance has been extensively studied and established in several species [1]. Increasing the IMF content modifies the fatty acid composition of the meat, affecting its nutritional quality [2]. High dietary intake of fat and saturated fatty acids (SFA) is related to a higher risk of cardiovascular disease [3], while the intake of unsaturated fatty acids, both monounsaturated (MUFA) and polyunsaturated (PUFA), has beneficial effects [4]. Nowadays, consumers are more aware of the importance of a healthy diet and demand quality products. Higher IMF content is desired to improve the juiciness, tenderness, and flavor of meat [1]; however, achieving this without decreasing the nutritional value would be optimal. For that reason, it is essential to understand the metabolism involved in fat deposition and its fatty acid composition.

IMF can be easily modified through changes in the diet [5,6] and also through selection due to its high heritability and adequate variability [7,8,9]. A few selection experiments have been carried out in species including cattle [8], chicken [9] and pigs [7], with successful results. In rabbits, a divergent selection experiment for IMF content in the *Longissimus thoracis et lumborum* (*LTL*) muscle was performed at the Universitat Politècnica de València [10]. The experiment led to the creation of two rabbit lines, one with high-IMF content (H) and the other with low-IMF content (L). The divergent lines were contemporarily reared under the same environmental conditions, and they were fed the same diet. Hence, the differences found between them and in their metabolism can be directly attributed to their genetic composition. These lines constitute a very valuable material to study the genetic effect of the selected and correlated traits, such as fatty acid composition and carcass fat deposition.

Liver metabolism is essential in fat deposition because of its central role in the transformation and distribution of the nutrients ingested in the diet [11], and in lipid metabolism and transport [12]. The intestine and its microbiota composition play a central role in the digestion of nutrients in rabbits [13]. Once absorbed, the nutrients are transported to the liver, where the hepatocytes transform them into the fuels and precursors required by other tissues [11]. The fatty acids that enter the hepatocytes after digestion are used for the biosynthesis of lipids that can be either transported to the muscles and adipose tissue or stored in the liver [11]. The study of liver lipid deposition and its fatty acid content and comparison with that of the muscle could help understand the mechanisms of lipid metabolism and transport employed by the liver. The metabolic activity of the liver also leaves traces that can be tracked by measuring some plasma metabolic markers, and its analysis will cast more light on the metabolic routes involved.

The aim of this study was to estimate the direct response to selection for IMF after 10 generations and the correlated responses in liver fat, liver fatty acid content, and plasma metabolic markers related to liver metabolism. The correlated responses in carcass quality traits and meat fatty acid content were also studied. The analysis of these results and their relationship will help in understanding the metabolism of fat deposition.

## 2. Materials and Methods

### 2.1. Animal Material

The samples for this study were taken from two rabbit lines divergently selected for intramuscular fat content (IMF) in the *Longissimus thoracis et lumborum* (*LTL*) muscle. Rabbits from the 9th and 10th generations of selection were used in this study. All samples were taken at 9 weeks of age, and the average weight of the animals was 1.9 kg.

The divergent selection started from a base population with 13 sires and 83 does. From generations 1 to 7, the high-IMF (H) and low-IMF (L) lines had 8 sires and 40 does each, increasing them to 10 sires and 60 does from the 8th generation forward. The animals were contemporaneous in each generation, and they were reared under the same environmental conditions. The selection criterion was the IMF content measured at 9 weeks of age in two young rabbits (one male and one female) of each doe. The young rabbits were weaned at 28 days, housed collectively, and fed ad libitum with a standard commercial diet until slaughter.

Each sire was mated with six does. The dams were ranked according to the average IMF values obtained from their offspring, and the top 20% of dams provided all females for the next generation. To reduce inbreeding, only one male progeny born from the highest-ranked mate of each sire was selected for the next generation. The management and complete selection procedure is described in Martínez-Álvaro et al. (2016) [10].

### 2.2. Sampling

#### 2.2.1. Meat and Carcass Quality Traits

The direct and correlated responses to selection in meat and carcass quality traits in the 10th generation were studied using 190 rabbits, 101 from the H line (60 males and 41 females) and 89 from the L line (42 males and 47 females). In this generation, 82 out of the 190 rabbits were involved in a study of maternal effects and were born after an embryo transfer procedure. The commercial diet had an average composition of 16% of crude protein, 2.4% of fat, and 16.5% of crude fiber. The fatty acid composition of the diet, expressed as a percentage of total fatty acids, was 16.6% C16:0, 0.5% C16:1, 1.8% C18:0, 19.7% C18:1, 54.8% C18:2, 5.7% C18:3 and 0.9% fatty acids with more than 20 carbon atoms (C > 20).

The animals were slaughtered at 9 weeks of age, after 4 h of fasting, by exsanguination after electrical stunning. Their carcasses were then chilled for 24 h at 4 °C. After refrigeration, the carcass weight was recorded as commercial carcass weight (CCW). The reference carcass weight (RCW) was recorded as the carcass weight without the head, liver, kidney, and the organs of the chest and neck, according to the norms of the World Rabbit Science Association [14]. The liver (LW), scapular fat (SF), and perirenal fat (PF) were excised and weighed, and their percentages were calculated relative to the RCW. The carcass total fat content was measured with the dissectible fat percentage (DF), calculated as the sum of SF and PF percentages.

For the IMF quantification, the *LTL* muscle was excised, minced and lyophilized. The lyophilized powder was scanned using near-infrared spectroscopy (NIRS, 5000, FOSS NIRSystems Inc., Hilleroed, Denmark) and intramuscular fat content was estimated as g IMF/100 g of fresh muscle, applying the equations developed by Zomeño et al. [15].

#### 2.2.2. Meat Fatty Acid Content

The differences in meat fatty acid content between lines were estimated using 473 samples of rabbits from the 9th generation: 237 from the H line (118 males and 119 females) and 236 from the L line (115 males and 121 females). Animals were slaughtered and the carcasses followed the same procedures described above. The IMF content of the *LTL* muscle was also determined by NIRS.

For the quantification of fatty acids (FA), the fatty acid methyl esters (FAME) of each lyophilized *LTL* sample were prepared following the method described in O’Fallon et al. (2007) [16]. The FAME were analyzed using gas chromatography (FOCUS, Thermo, Milan, Italy), with split/splitless injection, flame ionization detection, and a fused silica capillary column SPTM 2560 (Supelco, Bellefonte, PA, USA) (100 m × 0.25 mm × 0.2 µm film thickness). The samples were injected using a split ratio of 1/100 and helium as a carrier gas at a constant flow-rate of 1 mL/min and a linear velocity of 20 cm/s. The oven temperature was set at 140 °C for 5 min, then was increased to 240 °C at a rate of 4 °C/min and finally maintained at that temperature for 30 min. The detector and the injector were at 260 °C. The fatty acids were identified by comparing their retention time with the standards supplied by Supelco (Bellefonte, PA, USA), and quantified using C13:0 as an internal standard. Taking into consideration that the absolute amount of fatty acid increases together with the amount of lipid, each fatty acid was expressed as a proportion of 100 mg of total lipids present in the sample (mg FA/100 mg IMF).

The individual fatty acids studied were C14:0, C15:0, C16:0, C17:0, C18:0, C16:1n7, C17:1n10, C18:1n7, C18:1n9c, C18:1n9t, C20:1n11, C18:2n6, C18:3n3, C20:2n6, C20:3n6, C20:4n6, C22:4n6, C22:5n3, and C22:6n3. The saturated (SFA), monounsaturated (MUFA) and polyunsaturated fatty acids (PUFA), the n3 and n6 groups, and the ratios PUFA/SFA and MUFA/SFA were also analyzed. Other minor individual fatty acids such as C12:0, C18:3n6, C20:0, C20:3n3, C20:5n3, C22:0, and C22:1n9 were not studied individually; however, they were considered when estimating SFA, MUFA, and PUFA groups.

#### 2.2.3. Liver Fat, Liver Fatty Acid Content, and Plasma Metabolic Markers

A random subsample of 27 rabbits from the H line (13 males and 14 females) and 27 from the L line (13 males and 14 females) from the 10th generation was used to estimate the differences between lines in liver fat, liver fatty acid content, and plasma metabolic markers related to lipid metabolism.

The total lipid content of the liver was determined by ether extraction (Soxtec 2055, Tecator, Höganäs, Sweden) with previous acid hydrolysis (Soxcap 2022, Tecator, Höganäs, Sweden). Three replicates per sample of liver tissue were analyzed. The average lipid content of the replicates was calculated, and the results were expressed as g lipids/100 g of liver.

For the quantification of fatty acids, the FAME of each liver tissue sample were prepared, followed by their analysis using gas chromatography (FOCUS, Thermo, Milan, Italy), as described earlier. Each fatty acid was expressed as mg FA/100 mg of lipid content.

The individual fatty acids studied were C14:0, C15:0, C16:0, C17:0, C18:0, C16:1n7, C17:1n10, C18:1n7, C18:1n9c, C18:1n9t, C18:2n6, C18:3n3, C18:3n6, C20:2n6, C20:3n6, C20:3n3, C20:4n6, C22:4n6, C22:5n3, and C22:6n3. The groups SFA, MUFA, PUFA, n3, and n6, and the ratios PUFA/SFA and MUFA/SFA were also analyzed. Other minor individual fatty acids such as C12:0, C20:0, C22:0, C24:0, C20:1n11, C20:5n3, and C22:1n9 were not studied individually; however, they were considered when estimating SFA, MUFA, and PUFA groups.

To obtain the plasma metabolite concentrations, blood samples were collected at slaughter from the jugular vein in 1 mL lyophilized lithium heparin tubes. The plasma was separated by centrifugation at 3000 rpm for 10 min. Plasma concentrations of glucose, triglycerides, cholesterol, bilirubin, albumin, bile acids, and alanine transaminase (ALT) were determined in the laboratory of the Veterinary Clinical Hospital of the CEU-Cardenal Herrera University. All the parameters considered in this study were analyzed using an automatic clinical chemistry analyzer Spin 200E, using reagents from the manufacturer (Spinreact, Girona, Spain). Plasma concentrations of glucose, triglycerides, and total cholesterol were determined by enzymatic colorimetric methods. Glucose was determined by the Trinder’s glucose oxidase method [17], triglycerides by the glycerol phosphate dehydrogenase-peroxidase method [18], and total cholesterol by the cholesterol oxidase-peroxidase method [19]. Plasma concentrations of bilirubin and albumin were determined by colorimetric methods: the first one using dimethyl sulfoxide [20] and the second one using bromocresol green [21]. Finally, the bile acids concentration was determined using an enzymatic-colorimetric method [22], and the ALT using a photometric method [23].

### 2.3. Statistical Analysis

Descriptive statistics of all studied traits were estimated after correcting the data by the fixed effects of line (high or low), sex (male or female) and parity order (2 levels: first parity or second and more). Additionally, the fixed effect of month was included in the model for meat fatty acid content (5 levels: 1 for each month in which the samples were taken), and the fixed effect of month-transference (3 levels) in the model for meat and carcass quality traits. The month-transference effect classifies the samples that were taken in two different time periods and that were born after an embryo transfer procedure. The model for the liver fatty acid content and the plasma metabolites did not include a month (or month-transference) effect because all samples were taken within the same period and there was no embryo transfer procedure involved.

The differences between lines were estimated as the phenotypic differences between high- and low-IMF lines. The model was
***y*** = ***Xb*** + ***Wc*** + ***e***
where ***y*** is the vector of phenotypes, ***b*** is the vector of fixed effects including the line, sex, parity order, and month or month-transference (where applicable), ***c*** is the vector of common litter random effect, and ***e*** is the vector of residual variance. ***X*** and ***W*** are the incidence matrices.

Bayesian inference was applied [24,25] using the program Rabbit (Institute for Animal Science and Technology, UPV, Valencia, Spain). Bounded flat priors were assumed for all fixed effects and variances. The common litter (***c***) and the residual (***e***) random effects were assumed to be uncorrelated and a priori distributed as follows
***c****~* N (0, **I**σ^2^_c_)
***e****~* N (0, **I**σ^2^_e_)

The marginal posterior distributions of the phenotypic differences between high- and low-IMF lines were obtained by Gibbs sampling. The parameters of the posterior distributions taken into consideration were the median of the difference (D_H-L_), the highest posterior density interval at 95% (HPD_95%_), and the probability of the difference being greater than zero when D_H-L_ > 0, or lower than zero when D_H-L_ < 0 (P_0_). In addition, relevant values of the differences (r) were proposed as 1/3 of the phenotypic standard deviation of the trait [25]. The probability of the difference (D_H-L_) being greater than r when D_H-L_ > 0, or lower than −r when D_H-L_ < 0 was calculated (probability of relevance, P_r_).

## 3. Results and Discussion

### 3.1. Direct and Correlated Response to Selection after 10 Generations of Selection

The analyses performed by Martínez-Álvaro et al. (2016) [10] proved that the direct and correlated responses to selection in this experiment were symmetrical. Therefore, the positive phenotypic differences between H and L (D_H-L_) would mean an increase in the H line and a decrease in the L line, while a negative difference would mean the opposite.

Table 1 shows the results for the intramuscular fat (IMF) and carcass traits. The divergent selection for IMF was successful. The response to selection in the 10th generation (D_H-L_) was 0.49 g IMF/100 g of *LTL* muscle, equivalent to 3.8 SD of the trait. The difference was relevant (P_r_ = 1). This response is in agreement with those reported in previous generations [10,26], increasing at an average rate of 5% of the mean per generation. The divergent lines originated from the same base population, were contemporarily reared under the same environmental conditions, and were fed the same commercial diet. As both lines were under the same environmental conditions, the differences found between them can be directly attributed to their genetic composition.

Selection for IMF content did not have an impact on carcass weight (CCW; D_H-L_ = −0.36 g), but it did lead to a greater dissectible fat percentage (DF) in the H than in the L line (D_H-L_ = 0.63%, P_r_ = 1), which could impair carcass quality. However, in rabbits, this increase in DF would not have a relevant economic impact due to the low total DF of the carcass in this species. In addition, the moderate, positive genetic correlation between IMF and DF in these rabbit lines (0.34 [0.08 0.60]) [10] indicates that it would be possible to obtain an optimal profit including both traits in a selection index optimizing their economic weights.

### 3.2. Correlated Response in the Meat Fatty Acid Content

Selection for IMF modified the meat fatty acid content. Table 2 shows the descriptive statistics and the differences between lines in the fatty acid content of *LTL* muscle. The values are expressed in g/100 g IMF.

The SFA and PUFA groups constitute the most abundant fatty acids (26.46 g SFA/100 g IMF and 28.32 g PUFA/100 g IMF), while the MUFA group was the least abundant group (18.14 g MUFA/100 g IMF). Regarding the individual fatty acids, the linoleic (C18:2n6c; 20.12 g/100 g IMF), palmitic (C16:0; 18.71 g/100 g IMF) and oleic acids (C18:1n9c; 17.89 g/100 g IMF) were the most abundant, followed by stearic (C18:0; 5.97 g/100 g IMF) and arachidonic acids (C20:4n6; 5.23 g/100 g IMF). PUFAs in rabbit meat are characterized by their high content of n6, mainly due to the high content of C18:2 [27], and by their low content of n3 (26.78 g for n6 and 1.53 g for n3).

Increasing the IMF content of the H line led to a relevant increase in the SFA (D_H-L_ = 5.05 g SFA/100 g IMF, Pr = 1) and MUFA groups (D_H-L_ = 5.04 g MUFA/100 g IMF, P_r_ = 1). The PUFA group was greater in the H than in the L line, but the difference was not relevant (P_r_ = 0.05). The variation in the fatty acid profile caused by differences in the fat content has been previously reported [1,21]. Higher SFA and MUFA percentages are expected as muscle lipids increase because the triglycerides fraction, rich in SFA and MUFA, increases, whereas the phospholipids fraction, rich in PUFA, increases at a slower rate [28].

In general, individual SFA and MUFA showed similar patterns to their groups, as they were greater in the H than in the L line and showed relevant differences. The stearic acid (C18:0) was the only SFA higher in L line (D_H-L_ = −0.81 g FA/100 g IMF). This could be because, in the *LTL* muscle of rabbits, the triglycerides have a lower proportion of C18:0 than phospholipids [29], a characteristic not observed in other species including pigs, lambs, and cattle [30]. All the individual MUFA were higher in the H line save for C17:1, which showed no relevant differences between lines.

The ratio between C18:1n9 and C18:0 is often used as an indicator of the stearoyl-CoA desaturase activity. In these lines, the ratio was greater in the H than in the L line, and the difference was relevant (D_H-L_ = 0.98, P_r_ = 1). A GWAS previously performed on the *LTL* fatty acid content of these rabbit lines identified the stearoyl-CoA desaturase 1 (SCD) as an important gene related to the C18:0 and C18:1 content [31]. This gene encodes the enzyme that catalyzes the formation of MUFA from the SFA. Together, these findings indicate the higher activity of the SCD in the H line, which can relate directly to the different genetic composition of the lines.

Concerning the individual PUFA, the linoleic (C18:2n6) and α-linolenic (C18:3n3) acids were greater in the H than in the L line. The triglycerides and phospholipids fractions in the *LTL* muscle of rabbits have similar percentages of C18:2n6 [29]. However, this fatty acid accounts for around 80% of the total PUFA in triglycerides, while in phospholipids this figure is around 54%. The higher amount of triglycerides in the H line, together with its higher proportion of PUFA, could explain this difference. Triglycerides also have a greater percentage of C18:3n3 than phospholipids [29], and animals with higher fat content will have a higher abundance of this fatty acid. All the remaining individual PUFA (C20:2n6, C20:3n6, C20:4n6, C20:5n3, C22:4n6, C22:5n3, and C22:6n3) were more abundant in the L than in the H line. Unsaturated fatty acids longer than 18 carbons are not detected in the triglyceride fraction of rabbit meat, except for a small amount of C20:1 [32], thus explaining this difference regarding C18:2n6 and C18:3n3.

Relevant correlated responses were obtained for the ratios PUFA/SFA and MUFA/SFA. The MUFA/SFA ratio was higher in the H line (D_H-L_ = 0.17), while the PUFA/SFA ratio was higher in the L line (D_H-L_ = −0.19). The indices analyzed are good indicators of the nutritional quality of the meat. High SFA intake has been related to coronary disease [3], whereas unsaturated fat, especially PUFA, has beneficial effects. Recommendations state that ratio PUFA/SFA the should be at least 0.6 [33]. Despite the differences between lines, both meet this requirement. The total amount of intramuscular fat should also be considered in analyzing the nutritional quality of the meat. However, although there was an important response to selection, the total IMF content in both lines is low (H = 1.3 and L = 0.8 g IMF/100 g of *LTL*) and would not constitute a nutritional disadvantage.

### 3.3. Correlated Response in Liver Fat and Plasma Metabolites

The liver is central to the organism’s overall metabolism, including fatty acid metabolism and lipid circulation via lipoprotein synthesis [11]. The liver is also the main lipogenic site in growing rabbits [34]. Hence, the study of its activity is essential to understand the metabolism of fat deposition. Table 3 summarizes the results of the analysis of liver traits and plasma metabolites related to liver metabolism.

The H line showed a greater liver size than the L line with a relevant difference (D_H-L_ = 0.88%, P_r_ = 1), which has been related to higher lipogenic activity in a previous generation [35]. However, despite the higher lipogenic activity, no difference between lines was found in the liver lipid content at 9 weeks of age (D_H-L_ = −0.04 g lipid/100 g liver, P_0_ = 0.62). Triglycerides may accumulate in the hepatocytes or be transported as constituents of very low density lipoproteins (VLDL) [11]. Abnormal lipid accumulation within hepatocytes (i.e., hepatic steatosis) can occur under specific situations, such as obesity [36] or genetic defects [37]. These results indicate that higher fat deposition observed in the H line did not influence lipid deposition in the liver.

All plasma metabolites measured in these rabbit lines had normal concentrations (Table 3) [38,39]. The plasma concentrations of triglycerides (P_0_ = 1), cholesterol (P_0_ = 0.74) and bile acids (P_0_ = 0.80) were greater in the L than in the H line, but the difference was only relevant for triglycerides (P_r_ = 0.92). No differences were found in glucose (P_0_ = 0.53), bilirubin (P_0_ = 0.59) and ALT (P_0_ = 0.63) concentrations. Finally, the concentration of albumin was higher in the H line (P_0_ = 0.90), but the difference was not relevant (P_r_ = 0.67).

The plasma concentrations of glucose, triglycerides, cholesterol, and bile acids are useful indicators of the metabolism of carbohydrates and lipids. In rabbits, the glucose plasma level remains constant because they eat throughout the day [40] and use volatile fatty acids produced by cecal flora as their primary energy source [41]. The glucose plasma level is maintained even after a prolonged fast, probably due to cecotrophy [38].

Overall, the greater concentration of plasma lipids found in the L compared to the H line, together with its lower IMF and DF content, could be partially explained by a reduced uptake in their muscle and adipose tissue. The APOLD1 gene, involved in lipid-binding, transportation, and localization, was associated with the IMF trait [26] in a previous study performed in the 9th generation of selection of these lines, supporting the hypothesis. Additionally, a study performed in rats showed that the serum triglyceride-rich lipoproteins act as an inhibitor of the fatty acid synthesis in the hepatocytes [42], which is in line with the lower liver lipogenic activity previously observed in the L line [35]. The bile acids are synthesized from cholesterol and play an important role in the processing of dietary fats. Persistently raised bile acids have been reported in association with hepatic disease [41], but normal values were found in both lines and any differences between them were not relevant.

Finally, albumin, the most abundant circulating protein, is synthesized in the liver and is responsible for the transport of bilirubin and free fatty acids in the plasma. The higher concentration of albumin in the H line could mean a higher transport flux of free fatty acids in this line.

### 3.4. Correlated Response in the Liver Fatty Acid Content

Selection for IMF modified the liver fatty acid profile. Descriptive statistics and differences between lines are shown in Table 4. The values are shown in g FA/100 g lipids.

The SFA and PUFA groups constitute the most abundant groups (29.02 g SFA/100 g lipids and 29.96 g PUFA/100 g lipids), while the MUFA group is the least abundant (11.69 g FA/100 g lipids). A lower content of MUFA was observed in the liver, compared to the *LTL* muscle (18.14 g FA/100 g IMF), which appears to be mainly due to the lower proportion of oleic acid (C18:1n9c) in the liver. Greater stearic acid content (C18:0) was also found in the liver than in the *LTL* muscle. It has been reported that PUFA inhibit the activity of stearoyl-CoA-desaturase (SCD), the enzyme that synthesizes C18:1n9c from C18:0 [43]. As mentioned earlier, the liver has first access to ingested nutrients and is responsible for their transformation and distribution [11]. The high PUFA content in the diet and, consequently, in the liver could explain the inhibition of SCD, the accumulation of C18:0, and the reduction in MUFA content observed. In addition, the main role of the liver in the lipogenesis of growing rabbits could offer an explanation for the difference found with the fatty acid content of *LTL*, as also observed by other authors [44,45].

The increase in IMF content led to an increase in MUFA (D_H-L_ = 1.46 g/100 g lipids, P_0_ = 0.97) and a decrease in PUFA groups (D_H-L_ = −1.46 g/100 g lipids, P_0_ = 0.90), although the difference was only relevant for MUFA (P_r_ = 0.86). No relevant differences were found for the SFA group (D_H-L_ = 0.69 g/100 g lipids, P_r_ = 0.40). The differences found were smaller in the liver than in the meat, which can be explained by the lack of difference between lines in the liver lipid content.

Overall, individual MUFA and PUFA showed similar patterns to their group. All the individual MUFA were more abundant in the H line (P_0_ ≥ 0.89), save for C20:1, which showed no differences between lines. All the differences were relevant except for C18:1n9t. As for the individual PUFA, relevant differences were found only for the linoleic (C18:2n6), α-linolenic (C18:3n3) and n3 docosapentanoic (C22:5n3) acids, all of which were more abundant in the L than in the H line. Opposite results were obtained for the C18:2n6 and C18:3n3 in the *LTL* muscle. However, the higher amount of triglycerides in the *LTL* muscle of the H line, together with a higher proportion of C18:2n6 and C18:3n3 in that fraction, could explain this inverted relationship.

In reference to the individual SFA, the pentadecanoic (C15:0) and heptadecanoic (C17:0) acids were both greater in the L than in the H line, with relevant differences (Table 4). The hepatic synthesis of odd-chain saturated fatty acids (OCFA) is dependent on the propionate yielded by intestinal bacterial fermentation of indigestible carbohydrates (dietary fibers) [46]. In these lines, higher propionate synthesis in the cecum of the L line was found [47]. The propionate also acts as an inhibitor of de novo lipogenesis in hepatocytes, by inhibiting the uptake of acetate [48,49], which could also help explain the lower lipogenic activity previously found in the L line.

The n6 and n3 fatty acids were greater in the L than in the H line, but the difference was only relevant for the n3 (Table 4). These differences were opposite to those reported earlier for the *LTL* muscle. Relevant correlated responses were obtained for the ratios MUFA/SFA and PUFA/SFA. The MUFA/SFA ratio was higher in the livers of the H line (D_H-L_ = 0.05), while the PUFA/SFA ratio was higher in the livers of the L line (D_H-L_ = −0.06).

The fatty acid composition of the liver is closely related to the composition of the diet [50,51,52], because of its central role in the transformation and distribution of the nutrients ingested [11]. Since these lines were fed the same diet, the differences found in the fatty acid content of the liver reflect their different utilization of nutrients. The former can relate directly to differences in their genome and microbiome composition.

## 4. Conclusions

Fat deposition is the result of a complex metabolic network, which can be easily influenced by many environmental factors. The relevance of this study lies in the possibility of directly relating the results obtained to the genetic composition of the lines. Selection for IMF content was successful, showing a direct response equivalent to 3.8 SD of the trait after 10 generations. Increasing the IMF content also led to higher fat deposition in the carcass, but not in the liver. The lower IMF and carcass fat content found in the L line, together with the higher concentration of lipids in the plasma of this line, could be partially explained by a reduced uptake in its muscle and adipose tissue.

The fatty acid contents of *LTL* muscle and liver were modified after selection. Modification of the fatty acid content of the *LTL* muscle can be related to the changes in the triglycerides and phospholipids fractions, due to the alteration in IMF content. However, alterations in the fatty acid content of the liver could be due to changes in the metabolic pathways of the liver itself and differences between the lines in the microbial activity during digestion.

This is the first study analyzing the changes in liver fat deposition and in its fatty acid content caused by selection for IMF content. These results suggested a common genetic regulation between fat deposition in muscle and carcass, which seems to be independent from that regulating fat deposition in liver. Nonetheless, there is a common genetic background regulating the fatty acid content of liver and *LTL* muscle, which can be associated with the fatty acid metabolism of the liver.

## Figures and Tables

**Table 1 animals-12-00893-t001:** Descriptive statistics and differences between lines of meat and carcass quality traits.

			Descriptive Statistics		Differences between Lines				
Trait ^1^	N_H_ ^2^	N_L_ ^2^	Mean	SD	D_H-L_ ^3^	HPD_95%_ ^4^	P_0_ ^5^	r ^6^	P_r_ ^7^
IMF (g/100 g *LTL*)	101	89	1.06	0.13	0.49	[0.44 0.53]	1	0.04	1
CCW (g)	101	89	1076	116.10	−0.36	[−36.64 33.54]	0.51	38.70	0.02
DF (%)	100	87	1.17	0.58	0.63	[0.45 0.81]	1	0.19	1

^1^ IMF: intramuscular fat; CCW: commercial carcass weight; DF: dissectible fat; ^2^ N_H_: number of samples from the H line; N_L_: number of samples from the L line; ^3^ median of the marginal posterior distribution of the differences between lines; ^4^ highest posterior density interval at 95% probability; ^5^ probability of the difference of being >0 (when positive difference) or <0 (when negative difference); ^6^ relevant value; ^7^ probability of the difference of being >r (when positive difference) or <−r (when negative difference).

**Table 2 animals-12-00893-t002:** Descriptive statistics and differences between lines of the meat fatty acid profile.

			Descriptive Statistics		Differences between Lines				
Muscle Fatty Acid Profile (g FA/100 g IMF)	N_H_ ^1^	N_L_ ^1^	Mean	SD	D_H-L_ ^2^	HPD_95%_ ^3^	P_0_ ^4^	r ^5^	P_r_ ^6^
C14:0	237	236	0.95	0.21	0.89	[0.84 0.94]	1	0.07	1
C15:0	237	236	0.32	0.03	0.06	[0.06 0.07]	1	0.01	1
C16:0	237	236	18.71	1.54	4.82	[4.47 5.15]	1	0.51	1
C17:0	237	234	0.38	0.05	0.02	[0.01 0.03]	1	0.02	0.60
C18:0	237	236	5.97	0.54	−0.81	[−0.92 −0.69]	1	0.18	1
SFA	237	234	26.46	1.84	5.05	[4.70 5.49]	1	0.61	1
C16:1	237	236	0.96	0.39	0.99	[0.90 1.08]	1	0.13	1
C17:1	237	236	0.11	0.08	−0.01	[−0.02 0.01]	0.78	0.03	0
C18:1n9t	236	234	0.06	0.02	0.02	[0.01 0.02]	1	0.01	1
C18:1n9c	237	236	14.89	1.28	3.75	[3.46 4.02]	1	0.43	1
C18:1n7	236	236	1.87	0.19	0.20	[0.17 0.24]	1	0.06	1
C20:1	237	236	0.16	0.03	0.06	[0.05 0.07]	1	0.01	1
MUFA	235	234	18.14	1.65	5.04	[4.69 5.43]	1	0.55	1
C18:2n6c	237	236	20.12	1.61	3.03	[2.69 3.37]	1	0.54	1
C18:3n3	237	236	0.87	0.13	0.56	[0.53 0.59]	1	0.04	1
C20:2n6	237	235	0.24	0.03	−0.02	[−0.03 −0.01]	1	0.01	1
C20:3n6	237	236	0.43	0.05	−0.14	[−0.15 −0.12]	1	0.02	1
C20:4n6	237	235	5.23	0.71	−2.49	[−2.64 −2.32]	1	0.24	1
C22:4n6	237	236	0.64	0.1	−0.25	[−0.27 −0.23]	1	0.03	1
C22:5n3	237	236	0.45	0.07	−0.22	[−0.24 −0.20]	1	0.02	1
C22:6n3	236	236	0.1	0.03	−0.04	[−0.05 −0.03]	1	0.01	1
PUFA	235	232	28.32	2.10	0.34	[−0.08 0.78]	0.94	0.70	0.05
n6	236	233	26.78	1.98	0.08	[−0.32 0.50]	0.66	0.66	0
n3	236	235	1.53	0.14	0.26	[0.23 0.29]	1	0.05	1
PUFA/SFA	235	230	1.08	0.07	−0.19	[−0.20 −0.17]	1	0.02	1
MUFA/SFA	235	232	0.68	0.03	0.17	[0.15 0.18]	1	0.01	1

^1^ N_H_: number of samples from the H line; N_L_: number of samples from the L line; ^2^ median of the marginal posterior distribution of the differences between lines; ^3^ highest posterior density interval at 95% probability; ^4^ probability of the difference of being >0 (when positive difference) or <0 (when negative difference); ^5^ relevant value; ^6^ probability of the difference of being >r (when positive difference) or <−r (when negative difference).

**Table 3 animals-12-00893-t003:** Descriptive statistics and differences between lines of liver traits and plasma metabolites related to liver metabolism.

			Descriptive Statistics		Differences between Lines				
Trait ^1^	N_H_ ^2^	N_L_ ^2^	Mean	SD	D_H−L_ ^3^	HPD_95%_ ^4^	P_0_ ^5^	r ^6^	P_r_ ^7^
Liver weight (%)	101	89	7.11	1.00	0.88	[0.57 1.18]	1	0.33	1
Liver lipid (g/100 g liver)	27	27	3.83	0.40	−0.04	[−0.31 0.20]	0.62	0.13	0.23
Glucose (mg/dL)	26	23	112.7	29.23	0.82	[−22 20.4]	0.53	9.74	0.19
Triglycerides (mg/dL)	25	24	79.91	30.55	−34	[−56.11 −9.82]	1	1.18	0.98
Cholesterol (mg/dL)	25	24	47.96	15.92	−3.85	[−16.43 7.98]	0.74	5.31	0.41
Bile acids (mol/mL)	26	24	15.74	7.10	−2.13	[−7.19 2.51]	0.80	2.37	0.46
Bilirubin (mg/dL)	26	24	0.19	0.06	0	[−0.04 0.03]	0.59	0.02	0.19
Albumin (g/dL)	26	24	3.08	0.59	0.3	[−0.17 0.74]	0.90	0.2	0.67
ALT (UI/l)	26	24	65.56	20.00	2.1	[−10.47 15.9]	0.63	6.67	0.24

^1^ ALT: alanine aminotransferase. ^2^ N_H_: number of samples from the H line; N_L_: number of samples from the L line; ^3^ median of the marginal posterior distribution of the differences between lines; ^4^ highest posterior density interval at 95% probability; ^5^ probability of the difference of being >0 (when positive difference) or <0 (when negative difference); ^6^ relevant value; ^7^ probability of the difference of being >r (when positive difference) or <−r (when negative difference).

**Table 4 animals-12-00893-t004:** Descriptive statistics and differences between lines of the liver fatty acid content.

			Descriptive Statistics		Differences between Lines				
Liver Fatty Acid Profile (g FA/100 g Lipids)	N_H_ ^1^	N_L_ ^1^	Mean	SD	D_H-L_ ^2^	HPD_95%_ ^3^	P_0_ ^4^	r ^5^	P_r_ ^6^
C14:0	25	27	0.32	0.10	0.04	[−0.03 0.11]	0.83	0.03	0.50
C15:0	26	27	0.20	0.03	−0.04	[−0.06 −0.01]	1	0.01	0.98
C16:0	26	27	14.16	1.85	0.45	[−0.87 1.69]	0.77	0.62	0.40
C17:0	26	27	0.63	0.12	−0.09	[−0.17 −0.02]	0.99	0.04	0.92
C18:0	26	27	13.62	1.60	0.24	[−0.75 1.29]	0.69	0.53	0.30
SFA	26	27	29.02	2.75	0.69	[−1.02 2.48]	0.77	0.92	0.40
C16:1	26	27	0.47	0.20	0.17	[0.04 0.32]	0.99	0.07	0.93
C17:1	26	27	0.13	0.03	0.02	[0 0.05]	0.97	0.01	0.87
C18:1n9t	26	25	0.08	0.03	0.01	[−0.01 0.03]	0.89	0.01	0.58
C18:1n9c	26	27	9.66	1.89	1.36	[−0.07 2.73]	0.97	0.63	0.86
C18:1n7	26	27	1.20	0.21	0.12	[−0.03 0.26]	0.95	0.07	0.73
C20:1	26	27	0.26	0.08	−0.01	[−0.05 0.04]	0.63	0.03	0.23
MUFA	25	27	11.69	2.01	1.46	[0.09 3.05]	0.97	0.67	0.86
C18:2n6c	26	27	20.64	2.60	−1.37	[−2.98 0.22]	0.96	0.87	0.74
C18:3n3	25	27	0.50	0.09	−0.08	[−0.14 −0.02]	1	0.03	0.95
C20:2n6	26	27	0.82	0.21	−0.09	[−0.22 0.03]	0.93	0.07	0.66
C20:3n6	26	27	0.78	0.15	0.03	[−0.06 0.12]	0.76	0.05	0.35
C20:4n6	26	27	5.75	0.84	0.10	[−0.47 0.64]	0.63	0.28	0.25
C22:4n6	26	27	0.79	0.14	0.01	[−0.08 0.10]	0.57	0.05	0.20
C22:5n3	25	27	0.30	0.05	−0.03	[−0.06 0.01]	0.94	0.02	0.71
C22:6n3	26	26	0.12	0.04	−0.01	[−0.03 0.01]	0.75	0.01	0.36
PUFA	26	27	29.96	3.65	−1.46	[−3.72 0.84]	0.90	−1.22	0.59
n6	26	27	28.85	3.55	−0.3	[−3.61 0.85]	0.88	1.18	0.55
n3	26	26	1.05	0.16	−0.1	[−0.2 0]	0.97	0.05	0.82
PUFA/SFA	25	27	1.04	0.09	−0.06	[−0.12 −0.01]	0.98	0.03	0.88
MUFA/SFA	26	27	0.41	0.06	0.05	[0 0.09]	0.98	0.02	0.90

^1^ number of samples; ^2^ median of the marginal posterior distribution of the differences between lines; ^3^ highest posterior density interval at 95% probability; ^4^ probability of the difference of being >0 (when positive difference) or <0 (when negative difference); ^5^ relevant value; ^6^ probability of the difference of being >r (when positive difference) or <−r (when negative difference).

## Data Availability

The datasets analyzed in this study are available under request.

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
