# Peer review of "Intramuscular Fat Selection in Rabbits Modifies the Fatty Acid Composition of Muscle and Liver Tissues"

_animals, 2022, doi:10.3390/ani12070893_

Round 1

Reviewer 1 Report

I consider this is a good article. with an innovative research topic, interesting and relevant to the rabbit farmers, since it marks some aspects of interest in the rabbit selection.

Comments on the article:

  1. Simple summary: They failed to define some abrevations that later defines.
  2. Keywords: It is recommended not to use words that are already in the title, in order to increase the chance of finding the article in a scientific journal.
  3. Results: The difference between lines were estimated as the phenotypic difference between high and low IMF lines, using a mixed effects model, which is appropiate, but in tables 2, 3 and 4, the descriptive statistics of the difference between lines are presented. I think it would have been more convenient for them  to put the significance of the differences, not just the differences mean, and the confidence interval, since some values go through zero and appear that there were differences, when they probably did not exist.

Reviewer 2 Report

In general, this work is really interesting and well written. My minor comments aim to increase the scientific soundness and clarity of it.

  1. Longissimus thoracis et lumborum (LTL) muscle„ - From anatomical point of view and according to Nomina Anatomica Veterinaria musculus longissimus lumborum and musculus longissimus thoracis are separate muscles. Of course, they both participate in formation of musculus longissimus (and one continues as the second one) but they are separate anatomical units.
  2. I do not understand why English names are mixed with Latin ones? Moreover I see no consequence. For example, if authors used “Longissimus thoracis et lumborum” then they should use name “hepar” for the liver. This should be clarified.
  3. Please provide any details of the experimental animals (Age, sex, weight etc.).
  4. In MM I see no information if this experiment was approved by any Ethical Committee.

Reviewer 3 Report

The work "Intramuscular fat selection in rabbits modifies the fatty acid composition of muscle and liver tissues" is interesting. Contains current topic. 
The manuscript requires changes to be made to make it easier for the reader to understand. 

Introduction 
"Higher IMF content is desired to improve the meat's juiciness, tenderness, and flavour" [please add reference]. 

Few selection experiments - please specify. 

I suggest editing the purpose of the work so that it is clearly formulated. Please also check whether the conclusions on page 11 relate directly to the purpose of the study. 

Results and discussion Table 1 I 2. Perhaps it is worth adding the difference in the number of samples (N) under the table. 

Statistical analysis (page 5). I suggest editing this passage so that it clearly describes the statistical methods. As they stand, these methods are unreadable. 

Linguistic proofreading of the article is required. 

Please pay attention to the selection of references for the work. Are the articles of the authors of this work included in the reference list essential for a citation? 

Round 2

Reviewer 3 Report

Reviewer thanks the authors for the changes made. The manuscript is much improved.